Charting the global footprint of borderline oxacillin-resistant Staphylococcus aureus (BORSA): the first systematic review and meta-analysis

Engku Abd Rahman Engku Nur Syafirah 1
Irekeola Ahmad Adebayo 1 2
Yamin Dina 3 4
Elmi Abdirahman Hussein 1 5
http://orcid.org/0000-0002-0034-7624 Chan Yean Yean 1 6 yeancyn@yahoo.com
1 Department of Medical Microbiology and Parasitology, School of Medical Sciences, Health Campus, Universiti Sains Malaysia , Kota Bharu, Kelantan , Malaysia
2 Department of Biological Sciences, Microbiology Unit, College of Natural and Applied Sciences, Summit University Offa , Offa, Kwara , Nigeria
3 Department of Veterinary Clinical Studies, Faculty of Veterinary Medicine, Universiti Malaysia Kelantan , Kota Bharu, Kelantan , Malaysia
4 Department of Clinical Laboratory Sciences, School of Science, University of Jordan , Amman , Jordan
5 Department of Medical Laboratory Sciences, Faculty of Medicine and Health Sciences, Jamhuriya University of Science and Technology , Mogadishu , Somalia
6 Health Campus, Hospital Universiti Sains Malaysia , Kota Bharu, Kelantan , Malaysia
Upadhyay Rohit
Electronic publication date: 2024 Dec 16
Publication date: 2024
Volume: 12
Electronic Location ID: e18604
Received 2024 Aug 26; Accepted 2024 Nov 7
Copyright: © 2024 Engku Abd Rahman et al.
Copyright year: 2024
Copyright holder: Engku Abd Rahman et al.
License: This is an open access article distributed under the terms of the Creative Commons Attribution License, which permits unrestricted use, distribution, reproduction and adaptation in any medium and for any purpose provided that it is properly attributed. For attribution, the original author(s), title, publication source (PeerJ) and either DOI or URL of the article must be cited.
License URL: https://creativecommons.org/licenses/by/4.0/

Keywords: Staphylococcus aureus, Borderline, Oxacillin-resistant, BORSA

Funding: The authors received no funding for this work.

==============================
Borderline oxacillin-resistant Staphylococcus aureus (BORSA) has been a persistent yet under-researched concern in the realm of antibiotic resistance, characterized by unique resistance mechanisms and potential for severe infections. This systematic review and meta-analysis consolidates data from 29 studies encompassing 18,781 samples, revealing a global BORSA prevalence of 6.6% (95% CI [4.0–10.7]). The highest prevalence was found in animals (46.3%), followed by food (8.9%), and humans (5.1%). Notably, significant regional disparities were observed, with Brazil exhibiting the highest prevalence at 70.0%, while The Netherlands reported just 0.5%. These findings underscore the multifaceted nature of BORSA epidemiology, influenced by local antibiotic usage practices and healthcare infrastructures. The analysis also reveals substantial heterogeneity (I2 = 96.802%), highlighting the need for improved reporting practices and tailored surveillance protocols that account for the specific contexts of each study. As antibiotic resistance continues to escalate, understanding BORSA’s global footprint is crucial for informing targeted interventions and optimizing antibiotic stewardship programs. This study fills critical gaps in current knowledge of BORSA and highlights the need for coordinated efforts among researchers, healthcare providers, and policymakers to develop effective strategies for addressing the rising threat of antibiotic-resistant pathogens like BORSA, including further exploration of its genetic and phenotypic characteristics.

Introduction

Over the past few decades, the rise and dissemination of antibiotic resistance in bacterial pathogens have posed significant public health challenges globally. Among these resistant organisms, Staphylococcus aureus, a versatile and resilient bacterium, has garnered particular attention. Of particular importance is the borderline oxacillin-resistant S. aureus (BORSA) which cannot be easily classified as fully methicillin-resistant S. aureus (MRSA) or methicillin-susceptible S. aureus (MSSA), often leading to misidentification that presents challenges in epidemiology and treatment (Hryniewicz & Garbacz, 2017). Unlike MRSA and MSSA which have a well-defined clinical presentations, BORSA strains exhibit resistance characteristics that complicate antibiotic selection and influence clinical outcomes.

BORSA strains are characterized by marginal resistance to penicillinase-resistant penicillin (PRPs), with oxacillin minimum inhibitory concentrations (MICs) typically ranging from 1–8 µg/ml (Krupa et al., 2015; Hryniewicz & Garbacz, 2017). In contrast to MRSA, BORSA does not possess the altered penicillin-binding protein (PBP2a) encoded by the mecA or mecC genes (García-Álvarez et al., 2011). Instead, resistance often stems from heightened beta-lactamase production or occasional mutations in PBP genes (Ba et al., 2019; Konstantinovski et al., 2021b). Notably, the hyperproduction of beta-lactamase, particularly in hospital strains, can lead to significant challenges in treatment, as this enzyme can degrade PRPs slowly, complicating antibiotic therapy (McDougal & Thornsberry, 1986).

Another characteristic of BORSA strains is the absence of species-specific proteins like thermonuclease or coagulase, complicating the identification of certain borderline-resistant strains within the broader category of S. aureus. A study reported an infection caused by a strain exhibiting borderline resistance to oxacillin and lacking both thermonuclease and coagulase–two fundamental taxonomic markers of S. aureus (Skinner et al., 2009). While identification of S. aureus is a common practice in clinical laboratories, the challenges primarily lie in accurately detecting and characterising the resistance mechanisms of BORSA strains, which can hinder effective antibiotic therapy (Skinner et al., 2009).

The borderline resistance of BORSA strains can lead to more severe clinical outcomes compared to infections caused by MSSA, primarily due to treatment challenges associated with their resistance profile (Hryniewicz & Garbacz, 2017; Konstantinovski et al., 2021b). Furthermore, treatment of severe BORSA infections may prove challenging, even with higher doses of oxacillin, as identifying S. aureus strains exhibiting borderline resistance can complicate the selection of appropriate antibiotic therapies (Skinner et al., 2009; Hryniewicz & Garbacz, 2017).

This systematic review and meta-analysis aims to aggregate the current global data on BORSA prevalence across diverse environments. By synthesizing data from various studies, including clinical, food-related, and animal sources, this review will elucidates BORSA prevalence alongside its geographic variability. Given the lack of a universally accepted definition for borderline oxacillin-resistant strains, including varying terminology such as “methicillin-resistant lacking mec (MRLM)”, this study underscores the importance of clarity in terminology as research on antibiotic resistance continues to evolve. Additionally, this comprehensive synthesis intends to highlight existing knowledge gaps and suggest areas for further research. Such insights are crucial as we work to mitigate the growing threat of antibiotic resistance and ensure effective treatment of staphylococcal infections worldwide.

Materials and Methods

Standards and study framework

Utilising the Preferred Reporting Items for Systematic Reviews and Meta-Analyses (PRISMA) guidelines (Moher et al., 2009; Engku Abd Rahman et al., 2022), a meta-analysis of documented cases of BORSA infection across the globe was conducted. The study protocol underwent submission to PROSPERO and obtained registration number CRD42024551780.

Eligibility criteria for included studies

The study encompassed the following categories of literature: (1) investigations detailing the prevalence of BORSA; defined as strains with oxacillin MICs ranging from 1–8 µg/ml (2) studies conducted within the past decade were incorporated to capture contemporary trends; (3) primary research, such as cross-sectional, cohort, and case-control studies conducted across diverse settings. Conversely, the following types of literature were excluded: (1) subjective pieces such as opinions, editorials, perspectives, book chapters, reviews, case reports, and data from websites; (2) studies where full texts were inaccessible; (3) investigations lacking clear or comprehensive data on BORSA prevalence; (4) studies reliant on self-reported cases rather than laboratory-confirmed diagnoses; (5) reports concerning oxacillin-sensitive S. aureus (OSSA), defined as strains that are fully susceptible to oxacillin, or oxacillin-resistant S. aureus (ORSA), which includes MRSA but not BORSA.

Literature search

To prevent duplication, a meticulous examination of records in the PROSPERO database and other electronic databases was conducted to ascertain the absence of ongoing or completed meta-analyses on the global prevalence of BORSA. Methodical investigations were performed throughout five electronic databases–PubMed, Google Scholar, Scopus, ScienceDirect, and Web of Science (Core Collection)–without restrictions on the timeframe of studies, language, or study design. A preliminary search was conducted on May 16th, 2024, followed by a final update search completed on July 2nd, 2024, yielding a total of 3,765 articles (Fig. 1).

Figure 1 Summary of PRISMA flow diagram of study selection.

The search approach employed a blend of relevant terms to examine the worldwide effect of BORSA infections. Boolean operators ‘AND’ and ‘OR’ were employed with predefined search terms including “borderline oxacillin-resistant Staphylococcus aureus”, “oxacillin-resistant”, and “BORSA” to ensure comprehensive coverage. Moreover, references and titles from the studies included were employed as additional search techniques. Comprehensive search strategies for each of the five data repositories are outlined in Table S1.

Duplicate studies were identified and excluded using Mendeley Desktop version 1.19.8 software (London, England, UK). Two authors meticulously reviewed the relevant articles, first by screening titles and abstracts, then by conducting a detailed assessment of the full-text articles. Discrepancies concerning article inclusion were addressed through discussion and consultation with two additional authors.

Data retrieval

The assessment of included studies involved scrutiny of their titles, abstracts, and full-texts. Data extraction was conducted using an Excel spreadsheet (Microsoft® Office, Bellingham,WA, USA) with predefined fields. Authors independently gathered the following details from qualifying studies: the surname of the lead author and publication year, the countries of origin for the samples (clinical, food, or animal), laboratory methods used for diagnosing BORSA, reported instances of BORSA infections and the total number of isolates tested, along with their respective proportions.

Quality assessment

Two authors individually evaluated the quality of selected studies using the Joanna Briggs Institute (JBI) assessment tool, which is specifically crafted for prevalence research (File S1) (Munn et al., 2015). This checklist assesses nine elements, including the suitability of the sampling frame, sampling method, sample size sufficiency, description of study participants and settings, adequacy of data analysis, use of reliable methods for the identified conditions, valid measurements for all participants, appropriate statistical methods, and a sufficient response rate. Each element was rated as “Yes”, “No”, “Unclear”, or “Not applicable”. A score of 1 was awarded for “Yes”, whereas “No” and “Unclear” were given a score of 0. The average score for each included study was then computed. The quality of the 29 studies was evaluated on a scale from one to nine (Table S2) and classified according to their overall score as “low quality” (<50%), “moderate quality” (50–70%), and “high quality” (>70%) (Ahmed et al., 2024).

Data integration and quantitative analysis

The DerSimonian-Laird approach was utilised to determine the global prevalence of BORSA, with subgroup analyses performed according to country and sample origins. Anticipating variability from the diverse locations and contexts of the studies, a random-effects model was applied.

Variation among studies was evaluated using the I2 statistics, with a value exceeding 75% indicating significant heterogeneity (Higgins & Thompson, 2002). Subgroup analysis by country and sample type (human, food, and animal) was carried out to derive regional prevalence estimates and assess factors contributing to variation. Publication bias was examined using a funnel plot, which displayed prevalence estimates against their respective standard errors. Egger’s test was used to evaluate asymmetry in the funnel plot, with a significance threshold set at <0.05. Sensitivity analysis was conducted to explore the influence of each study on the overall estimate. Data analysis and visualizations were performed using OpenMeta[Analyst] (version 10.12) and Comprehensive Meta-Analysis (CMA) (version 2.2.027) software (Irekeola et al., 2022; Engku Abd Rahman et al., 2022).

Results

Selection of the relevant studies

A comprehensive search across multiple databases initially identified 3,765 unique records. Following automatic deduplication, 2,033 articles were left for further screening based on predefined inclusion and exclusion criteria using their titles and abstracts. Of these, 2,004 articles were found irrelevant to the research objectives and were subsequently excluded. Ultimately, 29 articles met the criteria for inclusion in the systematic review and meta-analysis. The detailed selection process is illustrated in Fig. 1.

Features of the qualified studies

Among the 29 studies incorporated into the meta-analysis, which encompassed a total sample size of 18,781 samples, there were 576 documented cases of BORSA infection. Approximately 20.6% of these studies originated from the United States of America (USA), with data collected from a total of 19 countries worldwide. Samples were sourced from human (clinical), food, and animal. A variety of analysis methods were utilised, including antibiotic sensitivity testing (AST) techniques such as disk diffusion, broth dilution, agar dilution, E-test, antibiogram, and automated systems (e.g., VITEK, MicroScan). Additionally, characterisation and typing techniques were also employed to further confirm the BORSA strains including polymerase chain reaction (PCR), pulse field gel electrophoresis (PFGE), whole genome sequencing (WGS), multi-locus sequence typing (MLST), amplified fragment length polymorphism (AFLP), multiple locus variable number tandem repeat analysis (MLVA), surface plasmon resonance (SPR). Table 1 offers a comprehensive summary of the principal characteristics of the studies included in the analysis.

Table 1 Major characteristics of the qualified studies.

S/N	Author (year)	Study period	Location	Study design	Sample source	Analysis methods*	Total (samples/ isolates)	Positive cases (samples/ isolates)	Proportion (%)	
1	Al-Safaar & Al-Charrakh (2013)	2010–2013	Iraq	Cross-sectional	Clinical	AST	21	0	0	
2	Argudín et al. (2018)	2013–2015	Belgium	Retrospective	Clinical	AST	298	12	4	
3	Balslev et al. (2005)	2000	Denmark	Case-control	Clinical	AST, phage type, PFGE & genotyping	710	37	5.2	
4	Buchan & Ledeboer (2010)	NR	USA	NR	Clinical	AST	364	2	0.5	
5	Bystroń et al. (2010)	NR	France	NR	Food	AST, genotyping & MLST	132	8	6	
6	Dicko et al. (2023)	2014–2020	Mali	Retrospective	Clinical	AST	735	41	5.6	
7	Dillard et al. (1996)	1994	USA	Cross-sectional	Clinical	AST	252	27	10.7	
8	Huang, Yan & Wu (2000)	1990–1998	Taiwan	Cross-sectional	Clinical	AST	288	4	1.4	
9	Huang et al. (2018)	2001–2015	Taiwan	Retrospective cohort	Clinical	AST, MLST & PFGE	1,867	65	3.5	
10	Khorvash, Mostafavizadeh & Mobasherizadeh (2008)	2005–2006	Iran	Cross-sectional	Clinical	AST	90	23	25.5	
11	Konstantinovski et al. (2021a)	2018–2019	The Netherlands	NR	Clinical	AST, AFLP, cgMLST & WGS	204	5	2.5	
12	Konstantinovski et al. (2021b)	2014–2016	The Netherlands	Cross-sectional	Clinical	AFLP, MLST, MLVA, cgMLST, & wgSNP	8,345	8	0.1	
13	Krupa et al. (2014)	2013	Poland	NR	Food	AST & genotyping	263	22	8.4	
14	Krupa et al. (2015)	2011–2012	Poland	Cross-sectional	Food	AST	420	49	11.7	
15	Leahy et al. (2011)	1992–2007	USA	Retrospective	Clinical	AST	34	18	53	
16	Liu et al. (1990)	1985–1987	USA	Cross-sectional	Clinical	AST	88	61	69	
17	Ljiljana et al. (2008)	NR	Serbia	NR	Clinical	AST	402	16	4	
18	Maalej et al. (2012)	2006–2011	Tunisia	Cross-sectional	Clinical	AST & latex agglutination test	1,895	23	1.2	
19	Martineau et al. (2000)	NR	–	NR	Clinical	AST & Nitrofecin test	206	4	1.9	
20	Nakamura et al. (2002)	2001	USA	Cross-sectional	Clinical	AST & PFGE	122	2	1.6	
21	Perillo et al. (2012)	NR	Italy	NR	Food	AST & Nitrofecin test	39	2	5.1	
22	Sá-Leão et al. (2001)	1993–2000	Portugal	Cross-sectional	Clinical	AST, dot-blot hybridization, & MLST	1,001	8	0.8	
23	Santhosh et al. (2008)	NR	Malaysia	Cohort	Clinical	AST	37	10	27.02	
24	Santos et al. (2021)	NR	Brazil	Cross-sectional	Animal	MALDI-TOF, AST, PFGE & Rep-PCR	20	14	70	
25	Sawhney et al. (2022)	NR	USA	NR	Clinical	AST, MALDI-TOF, WGS, Beta lactamase activity, PBP2 LFD	102	33	32.4	
26	Sieber et al. (2011)	2005–2011	Switzerland	Cross-sectional	Animal	AST & genotyping	70	18	25.7	
27	Stańkowska et al. (2019)	NR	Poland	Retrospective	Clinical	AST & genotyping	249	12	4.8	
28	Tawil et al. (2013)	NR	Canada	NR	Clinical	SPR, PCR, DNA sequence analysis	250	27	10.8	
29	Zehra et al. (2020)	NR	India	NR	Food & Community	AST	277	25	9	
Notes:

* Antibiotic sensitivity testing (AST) consisting of disk diffusion, broth dilution, agar dilution, E-test, antibiogram, automated systems (e.g., VITEk, MicroScan). Meanwhile, polymerase chain reaction (PCR), pulse field gel electrophoresis (PFGE), whole genome sequencing (WGS), multi-locus sequencing typing (MLST), amplified fragment length polymorphism (AFLP), multiple locus variable number tandem repeat analysis (MLVA), surface plasmon resonance (SPR) are techniques used to characterise or typing BORSA strains.

Major outcomes of BORSA infection worldwide

Employing the random-effect model to derive the summary assessments, the combined prevalence estimate for BORSA infections globally was 6.6% (95% CI [4.0–10.7]) (Fig. 2). The findings indicated a high degree of variability (I2 = 96.802%, Q = 875.460; p < 0.001).

Figure 2 Forest plot of aggregated prevalence of BORSA detection worldwide (n = 29) (Liu et al., 1990; Dillard et al., 1996; Huang, Yan & Wu, 2000; Martineau et al., 2000; Sá-Leão et al., 2001; Nakamura et al., 2002; Balslev et al., 2005; Santhosh et al., 2008; Khorvash, Mostafavizadeh & Mobasherizadeh, 2008; Ljiljana et al., 2008; Buchan & Ledeboer, 2010; Bystroń et al., 2010; Leahy et al., 2011; Sieber et al., 2011; Maalej et al., 2012; Perillo et al., 2012; Al-Safaar & Al-Charrakh, 2013; Tawil et al., 2013; Krupa et al., 2014, 2015; Huang et al., 2018; Argudín et al., 2018; Stańkowska et al., 2019; Zehra et al., 2020; Konstantinovski et al., 2021a, 2021b; Santos et al., 2021; Sawhney et al., 2022; Dicko et al., 2023).

A subgroup meta-analysis was conducted to assess the prevalence of BORSA detection across various countries globally (Fig. 3). Data were available from 28 studies worldwide, with the USA (n = 6) representing the majority of these studies (Fig. 4; Table 2). Brazil exhibited the highest pooled prevalence estimate of 70.0% (95% CI [47.3–85.9]), whereas The Netherlands had the lowest estimate of 0.5% (95% CI [0.0–10.6]) (Fig. 4; Table 2). The Netherlands had the highest heterogeneity (I2 = 96.90%; p < 0.001), which may have influenced the overall variability.

Figure 3 Global distribution of BORSA cases reported.

Figure 4 Forest plot of sub-group analysis on prevalence of BORSA detection worldwide stratified by country (Liu et al., 1990; Dillard et al., 1996; Huang, Yan & Wu, 2000; Sá-Leão et al., 2001; Nakamura et al., 2002; Balslev et al., 2005; Santhosh et al., 2008; Khorvash, Mostafavizadeh & Mobasherizadeh, 2008; Ljiljana et al., 2008; Buchan & Ledeboer, 2010; Bystroń et al., 2010; Leahy et al., 2011; Sieber et al., 2011; Maalej et al., 2012; Perillo et al., 2012; Al-Safaar & Al-Charrakh, 2013; Tawil et al., 2013; Krupa et al., 2014, 2015; Huang et al., 2018; Argudín et al., 2018; Stańkowska et al., 2019; Zehra et al., 2020; Konstantinovski et al., 2021a, 2021b; Santos et al., 2021; Sawhney et al., 2022; Dicko et al., 2023).

Table 2 Subgroup analysis of global BORSA detection prevalence, categorised by country.

Subgroup	No of studies	Prevalence (%)	95% CI	I2 (%)	Q	Heterogeneity test	
DF	p	
Iraq	1	2.3	[0.1–27.7]	NA	NA	NA	NA	
Belgium	1	4.0	[2.3–7.0]	NA	NA	NA	NA	
Denmark	1	5.2	[3.8–7.1]	NA	NA	NA	NA	
USA	6	15.0	[4.4–40.3]	96.82	157.134	5	<0.001	
France	1	6.1	[3.1–11.7]	NA	NA	NA	NA	
Mali	1	5.6	[4.1–7.5]	NA	NA	NA	NA	
Taiwan	2	2.5	[1.0–5.9]	69.53	3.282	1	0.070	
Iran	1	25.6	[17.6–35.5]	NA	NA	NA	NA	
The Netherlands	2	0.5	[0.0–10.6]	96.90	32.292	1	<0.001	
Poland	3	8.2	[5.0–13.0]	77.09	8.729	2	0.013	
Serbia	1	4.0	[2.5–6.4]	NA	NA	NA	NA	
Tunisia	1	1.2	[0.8–1.8]	NA	NA	NA	NA	
Italy	1	5.1	[1.3–18.3]	NA	NA	NA	NA	
Portugal	1	0.8	[0.4–1.6]	NA	NA	NA	NA	
Malaysia	1	27.0	[15.2–43.3]	NA	NA	NA	NA	
Brazil	1	70.0	[47.3–85.9]	NA	NA	NA	NA	
Switzerland	1	25.7	[16.8–37.2]	NA	NA	NA	NA	
Canada	1	10.8	[7.5–15.3]	NA	NA	NA	NA	
India	1	9.0	[6.2–13.0]	NA	NA	NA	NA	
Overall	28	6.9	[4.2–11.3]	96.89	867.252	27	<0.001	
Notes:

CI, Confidence interval; I2, Heterogeneity; Q, Heterogeneity chi-square; df, Degree of freedom; p, p-value.

The total number of studies are as stated (n = 28/29) because one study collected samples from unknown country, and were thus, excluded. Analysis was conducted on data from a distinct source.

Another sub-group meta-analysis stratified according to sample sources for BORSA detection was also performed. The data was available for 28 studies around the world, with human (clinical) source (n = 22) representing the majority of the studies (Fig. 5; Table 3). Sources from animal exhibited the highest aggregated prevalence estimate of 46.3% (95% CI [11.7–84.8]), whereas sources from human (clinical) reported the lowest estimate of 5.1% (95% CI [2.7–9.4]) (Fig. 5; Table 3). However, sources from human (clinical) had the most heterogeneity (I2 = 97.32%; p < 0.001), which might have contributed to the overall variability in this study.

Figure 5 Forest plot of sub-group analysis on prevalence of BORSA detection worldwide stratified by source (Liu et al., 1990; Dillard et al., 1996; Huang, Yan & Wu, 2000; Martineau et al., 2000; Sá-Leão et al., 2001; Nakamura et al., 2002; Balslev et al., 2005; Santhosh et al., 2008; Khorvash, Mostafavizadeh & Mobasherizadeh, 2008; Ljiljana et al., 2008; Buchan & Ledeboer, 2010; Bystroń et al., 2010; Leahy et al., 2011; Sieber et al., 2011; Maalej et al., 2012; Perillo et al., 2012; Al-Safaar & Al-Charrakh, 2013; Tawil et al., 2013; Krupa et al., 2014, 2015; Huang et al., 2018; Argudín et al., 2018; Stańkowska et al., 2019; Konstantinovski et al., 2021a, 2021b; Santos et al., 2021; Sawhney et al., 2022; Dicko et al., 2023).

Table 3 Subgroup analysis of global BORSA detection prevalence, categorised by sample source.

Subgroup	No of studies	Prevalence (%)	95% CI	I2 (%)	Q	Heterogeneity test	
DF	p	
Clinical	22	5.1	[2.7–9.4]	97.32	782.430	21	<0.001	
Food	4	8.9	[6.4–12.3]	42.60	5.226	3	0.156	
Animal	2	46.3	[11.7–84.8]	91.41	11.637	1	<0.001	
Overall	28	6.5	[3.9–10.8]	96.91	874.762	27	<0.001	
Notes:

CI, Confidence interval; I2, Heterogeneity; Q, Heterogeneity chi-square; df, Degree of freedom; p, p-value.

The overall number of studies are as stated (n = 27/ 28) because one study collected samples from a combination of food and community, and were thus, excluded. Analysis was conducted on data from a distinct source.

Analyses of publication bias, quality assessment, and sensitivity

A funnel plot of all qualified studies was created to investigate publication bias. Visual inspection of the plot revealed asymmetry, indicating possible publication bias (Fig. 6). Nevertheless, Egger’s regression test for funnel plot asymmetry yielded a non-significant p-value of 0.75899.

Figure 6 Funnel plot illustrating publication bias in studies reporting the global prevalence of BORSA detection (Egger’s test: p = 0.75899).

The plot shows fewer studies on the right side compared to the left, resulting in observed asymmetry.

Notably, the studies included in the analysis exhibited high methodological quality (Table S2). By adhering to rigorous methodological standards, the risk of bias and inaccuracies in the analysis is minimised, ensuring that the findings accurately reflect the true prevalence of BORSA detection.

To evaluate the robustness of the prevalence estimates for BORSA detection, a sensitivity analysis was conducted to assess the effect of each qualified studies on the total prevalence aggregated. Excluding the Liu et al. (1990) study resulted in a prevalence value of 5.9% (95% CI [3.7–9.2]). A similar result was observed when Santos et al. (2021) study was excluded, yielding a prevalence estimate of 5.9% (95% CI [3.6–9.6]). These were the lowest value found (Fig. 7). Excluding the Konstantinovski et al. (2021a) study resulted in the highest prevalence value of 7.7% (95% CI [4.9–12.0]). Despite these variations in individual values, the overall prevalence estimates of BORSA detection worldwide remained stable across scenarios (Fig. 7).

Figure 7 Forest plot of sensitivity analysis on global prevalence of BORSA detection (Liu et al., 1990; Dillard et al., 1996; Huang, Yan & Wu, 2000; Martineau et al., 2000; Sá-Leão et al., 2001; Nakamura et al., 2002; Balslev et al., 2005; Santhosh et al., 2008; Khorvash, Mostafavizadeh & Mobasherizadeh, 2008; Ljiljana et al., 2008; Buchan & Ledeboer, 2010; Bystroń et al., 2010; Leahy et al., 2011; Sieber et al., 2011; Maalej et al., 2012; Perillo et al., 2012; Al-Safaar & Al-Charrakh, 2013; Tawil et al., 2013; Krupa et al., 2014, 2015; Huang et al., 2018; Argudín et al., 2018; Stańkowska et al., 2019; Zehra et al., 2020; Konstantinovski et al., 2021a, 2021b; Santos et al., 2021; Sawhney et al., 2022; Dicko et al., 2023).

Discussion

This meta-analysis provides novel insights into the global epidemiology of BORSA infections, highlighting a significant prevalence across diverse geographical regions which reflect the multifaceted nature of BORSA epidemiology and highlight the global relevance of this persistent yet under-researched public health issue. Furthermore, the diversity in analysis methodologies employed—from traditional culture-based techniques to advanced molecular methods—underscores the complexity in accurately identifying and characterizing BORSA strains in different epidemiological contexts.

The meta-analysis synthesized data from 29 studies involving 18,781 samples, identifying 576 cases of BORSA infection. The aggregated prevalence estimate of BORSA worldwide was 6.6% (95% CI [4.0–10.7]), indicating a noteworthy presence of these antibiotic-resistant strains in various settings. The distribution of studies revealed significant regional variation, with Brazil showing the highest prevalence estimate (70.0%) and The Netherlands the lowest (0.5%). This suggests that regional antimicrobial usage practices, healthcare infrastructure, and socio-economic factors may influence these variations. While a study by Dutra et al. (2021) highlights significant issues with antimicrobial use in pig farming in Brazil (Dutra et al., 2021), it is essential to note that the studies included in this meta-analysis do not specifically focus on pig populations. Therefore, while the high antimicrobial consumption in agriculture is a concern, its direct correlation with BORSA prevalence in humans or other non-animal settings remains to be established. This highlights the need for targeted interventions that consider local contexts to effectively address antibiotic resistance. Furthermore, analysis methods varied widely across studies, including susceptibility testing and typing methods such as PCR, WGS, and MLST. This diversity underscores the complexity and adaptability required in surveillance and diagnostic practices to accurately capture and monitor BORSA prevalence. To enhance the reliability of BORSA detection, we recommend that future studies adopt standardised methodologies that incorporate both conventional and molecular techniques. Such an approach would facilitate better comparisons across studies and contribute to more effective monitoring of antibiotic resistance.

The analysis detected substantial heterogeneity (I2 = 96.802%), attributed partly to differences in study locations, methodologies, and possibly variations in local antimicrobial resistance patterns. Subgroup analyses by country and sample source (human, food, and animal) further highlighted varying prevalence rates and heterogeneity levels across different contexts. Notably, studies originating from The Netherlands, despite being conducted in the same medical center, demonstrated high diversity in BORSA prevalence due to the specific populations studied; one focused on patients while another assessed carriage among healthcare workers during a potential outbreak. The observed heterogeneity across studies indicates a complex interplay of factors influencing BORSA prevalence, emphasising the need for further investigation rather than a straightforward call for standardised reporting and surveillance protocols. In terms of prevalence, the data indicate a high occurrence of BORSA in animals (46.3%; 95% CI 11.7–84.8), compared to humans (5.1%; 95% CI [2.7–9.4]) and food (8.9%; 95% CI [6.4–12.3]). This disparity can be attributed to several factors. Antibiotic use in veterinary medicine, particularly in agricultural settings, contributes to increased selective pressure for resistant strains (Van Boeckel et al., 2015). Furthermore, the close contact between humans and animals in farming and veterinary environments can facilitate the transmission of resistant bacteria (Pandey et al., 2024). Additionally, farms and animal habitats may harbour higher concentrations of resistant bacteria due to waste management practices and the unique microbiomes present in these environments (Larsson & Flach, 2022).

These findings highlight the critical need for enhanced collaboration between researchers, healthcare providers, and policymakers to implement effective infection control measures and optimise antibiotic stewardship programs in the fight against BORSA and other resistant pathogens. Effective antimicrobial stewardship practices, as outlined by Bankar et al. (2022), can prevent the spread of drug-resistant bacteria in healthcare settings and improve patient outcomes. In terms of study quality and potential biases, the funnel plot indicated asymmetry, suggesting possible publication bias. However, Egger’s regression test for funnel plot asymmetry yielded a non-significant p-value of 0.75899, indicating that any detected bias may not significantly affect the overall findings. Sensitivity analysis demonstrated that while the exclusion of specific studies could influence individual prevalence estimates, the overall global prevalence of BORSA remained relatively stable across scenarios, affirming the robustness of the findings.

Despite the insights gained from this meta-analysis, several knowledge gaps remain. There is a pressing need for research focused on the genetic and phenotypic diversity of BORSA strains to better understand their resistance mechanisms and epidemiology across different populations. Additionally, studies exploring the clinical outcomes associated with BORSA infections compared to those caused by other S. aureus strains would be beneficial. Future investigations should examine the effectiveness of antibiotic stewardship programs in reducing the prevalence of BORSA and the impact of antibiotic use on human health. Addressing these gaps is crucial for developing targeted interventions and improving public health responses to combat antibiotic resistance effectively.

Conclusion

In conclusion, this meta-analysis provides a comprehensive assessment of BORSA prevalence worldwide, highlighting significant regional disparities that underscore the need for tailored public health interventions and antimicrobial stewardship programs. Understanding these variations can inform policymakers about prioritizing resources and developing targeted strategies to combat antibiotic-resistant pathogens like BORSA, ultimately enhancing global health systems. To effectively address the challenges posed by BORSA, it is crucial to adopt standardized methodologies for its detection across studies. Continued monitoring and research are essential to mitigate the impact of BORSA and similar pathogens on public health.

Supplemental Information

Supplemental Information 1 Search strategy and keywords.

Supplemental Information 2 Quality assessment of included studies.

Supplemental Information 3 PRISMA checklist.

Additional Information and Declarations

Competing Interests

Author Contributions

Data Availability

The authors declare that they have no competing interests.

Engku Nur Syafirah Engku Abd Rahman conceived and designed the experiments, performed the experiments, analyzed the data, prepared figures and/or tables, authored or reviewed drafts of the article, and approved the final draft.

Ahmad Adebayo Irekeola performed the experiments, analyzed the data, prepared figures and/or tables, authored or reviewed drafts of the article, and approved the final draft.

Dina Yamin performed the experiments, analyzed the data, prepared figures and/or tables, and approved the final draft.

Abdirahman Hussein Elmi performed the experiments, analyzed the data, authored or reviewed drafts of the article, and approved the final draft.

Yean Yean Chan conceived and designed the experiments, performed the experiments, authored or reviewed drafts of the article, and approved the final draft.

The following information was supplied regarding data availability:

This is a systematic review/meta-analysis.

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
