# Peer review of "Charting the global footprint of borderline oxacillin-resistant Staphylococcus aureus (BORSA): the first systematic review and meta-analysis"

_PeerJ, doi:10.7717/peerj.18604_

## Round 0.1 · original submission · Major Revisions

The manuscript requires a thorough revision. The introduction should be restructured and include recent studies (as noted by Reviewer 1) to analyze and update the results. Please ensure that all references used for the meta-analysis are included, and consider providing a supplementary list of these studies.

Address the comments from both reviewers in a point-by-point manner. The study lacks detailed elaboration of facts and discussions on previous studies, and the paragraphs need to be better connected for the ease of readers.

Overall, this manuscript needs a major revision that incorporates the above comments and addresses the feedback from both reviewers.

Reviewer 1 ·

Basic reporting

The manuscript presents a systematic review and meta-analysis on the prevalence of BORSA.
Overall, the manuscript is well written: sufficient background has been provided; the structure of the manuscript (tables and figures) is adequate (I have comments on the tables, please see additional comments).

Experimental design

The authors describe how the systematic review and meta-analysis were carried out. Inclusion and exclusion criteria for studies reported in the literature were clearly described.

Validity of the findings

The study presents the overall prevalence of BORSA, including isolates from human, animal and environmental sources. It is important to highlight this information in the abstract. Furthermore, the authors highlighted the countries with the highest and lowest prevalence of BORSA. In Brazil, only 20 isolates of animal origin were analyzed, while in the Netherlands, there were 8345 isolates from human sources.

Additional comments

Line 132: please add italic to Staphylococcus aureus
Please review the prevalence of BORSA in the Netherlands. In Table 1 it is stated as 0.1% and in the abstract, 0.5%
In line 290, the authors comment on the heterogeneity of detection methods used in studies on BORSA. AST refers to determining the minimum inhibitory concentration? Please clarify in the table. How can PCR, WGS and MLST be used to detect BORSA? In Table 1, I suggest changing to analysis methods if these methodologies were used to characterize the isolates.
Based on the analysis of studies on BORSA, can the authors recommend methodologies that should be used to detect BORSA, in order to standardize studies? This could be included in the conclusion.
Was Table 3 constructed according to country data? Because 29 studies were included in the review. Please clarify in the table.

·

Basic reporting

General remarks:
Introduction
I believe the introduction would benefit from a more structured approach. Please structure your introduction more profound per paragraph, and decide upon what messages and background information is important to get across to your audience in each paragraph. In the current text you touch upon several topics in different paragraphs throughout the introduction.
Example: Resistance mechanisms are discussed in line 56/57 but also in 83-91, whilst the consequences for treatment in line 57-58; 80-81. I suggest to consolidate the resistance mechanisms in one paragraph and follow up with consequences for treatment, then you can also dedicate a paragraph to epidemiology and the difficulties that arise from a lack of a generally accepted definition.
Also some statements are being made that are not referenced, please make sure that all statements can be backed up by unambiguous references or rephrase it a little to stay on the cautious interpretation of previous findings. For example: could be; suggests; potentially etc etc. (Tip: The Manchester Academic phrasebank provides ample examples if you need some inspiration).
Point out knowledge gaps more clearly, as one of your stated aims is to identify knowledge gaps. Taking this into consideration I am surprised that the introduction mostly depends on older studies, since the publication of the 2017 review there have been some publications. It is up to the authors whether they decide to incorporate newer literature, but in my opinion it is recommendable.

Discussion
Here applies a similar comment as in the introduction, have a more structured approach, Make sure that conclusions and statements are backed up by sound arguments.

Experimental design

Results
Very thorough description of study results and a comprehensive overview of studies in BORSA prevalence.
Double check spelling and grammar.
I did not find a list of the included publications in the meta-analysis as a reference list. I wanted to do a random check to see if I could confirm the reported result, but could not find the first reference (AL-Safaar 2013) and didn’t have a list so that has not been done by me. Please add a reference list of the included study in the meta analysis.

Validity of the findings

Materials and Methods:
Provide the definition of BORSA that has been used in your study. This is important as there is no generally accepted definition.
Eligibility criteria for included studies are not completely clear to me. See the PDF.
Due to a lacking definition of BORSA, sometimes other authors use different terms as methicillin-resistant lacking mec (MRLM). Have studies using such been captured by the current approach?

Additional comments

Review
This is a systematic review upon a topic that receives very little attention among S.aureus literature, thus it may be challenging to collect enough good quality reports. However, the previous review upon this topic dates back to 2017 from Hryniewicz & Garbacz, and thus it would be very useful to see what kind of novel insights and epidemiologic data is available. Also it would be the first time someone systematically reviews the prevalence of BORSA. Therefore I highly encourage the authors to proceed with their report and I compliment them with the massive amount of work that was needed to review all literature.
However, before it is suitable for publications some revisions need to be made to improve structure and readability.

---

## Round 0.2 · accepted · Accept

Authors have addressed all of the reviewers' comments and this manuscript is ready for publication.

Reviewer 1 ·

Basic reporting

The manuscript presents a meta-analysis on the prevalence of BORSA. The review is well structured and presents relevant data on the topic.

Experimental design

Methods are described with sufficient details.

Validity of the findings

Results are clear, and conclusions are well stated.

Additional comments

Dear authors, thank you for all replies.